# Explaining the role of Intrinsic Dimensionality in Adversarial Training

**Enes Altinisik** [1]   **Safa Messaoud** [1]   **Husrev Taha Sencar** [1]   **Hassan Sajjad** [2]   **Sanjay Chawla** [1]

## Abstract

Adversarial Training (AT) impacts different architectures in distinct ways: vision models gain robustness but face reduced generalization, encoder-based models exhibit limited robustness improvements with minimal generalization loss, and recent work in latent-space adversarial training (LAT) demonstrates that decoder-based models achieve improved robustness by applying AT across multiple layers. We provide the first explanation for these trends by leveraging the manifold conjecture: off-manifold adversarial examples (AEs) enhance robustness, while on-manifold AEs improve generalization. We show that vision and decoder-based models exhibit low intrinsic dimensionality in earlier layers (favoring off-manifold AEs), whereas encoder-based models do so in later layers (favoring on-manifold AEs). Exploiting this property, we introduce SMAAT, which improves the scalability of AT for encoder-based models by perturbing the layer with the lowest intrinsic dimensionality. This reduces the projected gradient descent (PGD) chain length required for AE generation, cutting GPU time by 25–33% while significantly boosting robustness. We validate SMAAT across multiple tasks, including text generation, sentiment classification, safety filtering, and retrieval augmented generation setups, demonstrating superior robustness with comparable generalization to standard training.

## 1. Introduction

Adversarial Training (AT) is the most effective approach for improving the robustness of deep neural networks against small input perturbations (Bai et al., 2021; Kurakin et al.,

[1]Qatar Computing Research Institute, HBKU, Doha, Qatar [2]Faculty of Computer Science, Dalhousie University, Halifax, Canada. Correspondence to: Enes Altinisik <ealtinisik@hbku.edu.qa>.

*Proceedings of the 42nd International Conference on Machine Learning*, Vancouver, Canada. PMLR 267, 2025. Copyright 2025 by the author(s).

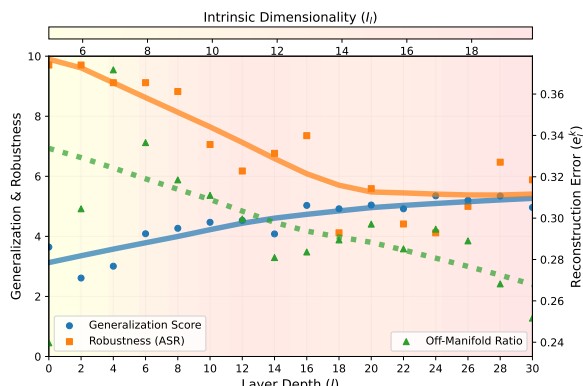

*Figure 1.* Impact of applying LAT at different layers of the LLaMA-2 model, illustrating the relationship between Intrinsic Dimensionality (background color), Generalization (blue), Robustness (orange), and Off-Manifold Ratio (green, based on reconstruction error). Markers show the average of measured values across multiple training configurations; lines depict overall trends. The off-manifold ratio measures the percentage of adversarial examples that fall outside the data manifold using reconstruction error. As we move to deeper layers, the Intrinsic Dimensionality increases, resulting in a decrease in the off-manifold ratio. According to the manifold conjecture, this leads to an increase in generalization (more on-manifold samples) and a decrease in robustness.

2017). It is formulated as a min-max optimization problem, where the outer minimization optimizes model parameters, and the inner maximization seeks worst-case input perturbations. In deep networks, the inner maximization is typically solved approximately using multiple iterations of projected gradient descent (PGD, Madry et al. (2018)). However, the trade-off between robustness and generalization remains poorly understood. For instance, AT reduces generalization in vision models, leading to an 8% drop on CIFAR-10 (Shafahi et al., 2019), whereas encoder-based language models often retain or even improve generalization, achieving a 1% gain on AGNEWS (Zhu et al., 2020). Moreover, robustness gains are significantly higher in vision models (e.g., 40% on CIFAR-10) compared to encoder-based models (e.g., 10% on AGNEWS). Recent work on Latent Adversarial Training (LAT) suggests that applying AT across multiple layers yields better robustness and generalization than focusing on a single layer (Sheshadri et al., 2024). While prior

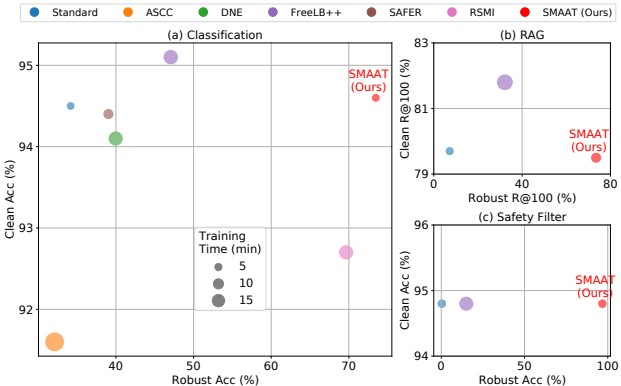

*Figure 2.* Comparison of SMAAT robustness (x-axis), generalization (y-axis), and run time (marker size) against baselines for robustifying (a) topic classifiers, (b) retriever models in the Retrieval Augmented Generation (RAG) setup and (c) safety filters for decoder-based LLMs. SMAAT significantly enhances model robustness compared to seven different baselines, while maintaining nearly the same clean accuracy. Besides, it is significantly more scalable than AT (marker size).

research has investigated generalization loss in vision models (Madry et al., 2018; Wang et al., 2019b; Altinisik et al., 2023a; Zhang et al., 2019b; Cheng et al., 2022), no study has systematically explored the robustness-generalization dynamics across vision models (CNNs and Vision Transformers), decoder-based (dec-LLMs), and encoder-based (enc-LLMs) language models. Additionally, the high computational cost of AT limits its practical deployment. Recent efforts to reduce the number of PGD steps—by reusing or accumulating gradients during updates (Shafahi et al., 2019; Zhang et al., 2019a; Zhu et al., 2020)—have improved efficiency but still require full network passes, making AT computationally expensive.

In this work, we investigate how differences in robustness and generalization trends across foundational models relate to the intrinsic dimensionality (ID) of the data manifold. The data manifold is a potentially non-linear subspace spanned by the dataset, and its dimensionality influences whether adversarial examples (AEs) lie on the manifold (on-manifold AEs) or fall outside it (off-manifold AEs) during training. We show that the ID of the data manifold in the first layer is much higher in enc-LLMs compared to vision models and dec-LLMs which results in AEs being more on-manifold (ONM-AEs) in enc-LLMs and more off-manifold (OFM-AEs) in vision models and dec-LLMs. In accordance with the manifold conjecture (Ethayarajh, 2019; Shamir et al., 2021; Gilmer et al., 2018), we find that OFM-AEs lead to better robustness, while more ONM-AEs lead to better generalization (Fig. 1, with details provided in Section 4). To the best of our knowledge, this is the first explanation for

the difference in the robustness magnitudes across vision models and enc-LLMs. Our findings are also consistent with YOPO (Zhang et al., 2019a) and TMD (Minh & Luu, 2022). Specifically, YOPO highlights the critical role of the first layer in vision models for AT, while TMD uses the last layer to detect AEs in enc-LLMs.

We further leverage our insights on the impact of the manifold ID on robustness and generalization and hypothesize that *perturbing the intermediate layer $l$ with the highest off-manifold AE ratio (equivalently lowest ID) should achieve high robustness at low computational cost.* Intuitively, a lower ID corresponds to a higher proportion of OFM-AEs generation and a greater improvement in robustness by the manifold conjecture. Additionally, layers closer to the output result in shorter PGD chains, leading to efficient AT. To this end, we propose SMAAT[1], a Scalable Manifold Aware AT approach that applies AT at the layer with the highest proportion of OFM-AEs. We found that this critical layer is consistently the **last** layer across enc-LLMs on several applications. Hence, for enc-LLMs, SMAAT leads to a significant speed-up of AT by avoiding a full backward pass as it calculates the gradients only until the last layer rather than the entire network. Moreover, we empirically show that this results in higher robustness. Yet, this ID trend is different for vision or dec-LLMs, where we find that the first layer is always the one with the lowest ID. In these cases, SMAAT effectively reduces to standard AT.

Given enc-LLMs continue to play a crucial role in machine learning pipelines, there is significant value in effectively enhancing their robustness. To this end, we evaluated SMAAT in improving the robustness of (1) classifiers, (2) safety filters, and (3) retrievers within Retrieval-Augmented Generation (RAG). SMAAT achieved state-of-the-art (SOTA) runtime and robustness results on all tasks, while maintaining clean accuracy (generalization) comparable to standard training. Specifically, in sentiment and content classification tasks, SMAAT improved the robustness of enc-LLMs on AG-NEWS (Zhang et al., 2015), IMDB (Maas et al., 2011), and YELP (Zhang et al., 2015) datasets by 8.6%, 15.7%, and 28.8% for BERT (Devlin et al., 2019) and by 6.0%, 5.8%, and 19.0% for RoBERTa (Liu et al., 2019), respectively. For safety filtering, SMAAT accurately identified harmful prompts generated by GCG (Zou et al., 2023), achieving 97-100% accuracy. In RAG experiments, SMAAT significantly enhanced the robustness of the Contrevier model (Izacard et al., 2022) on the RAG setup, achieving over 80% robustness against poisoning attacks (Zhong et al., 2023). Besides, SMAAT required only about 25-33% of the GPU time compared to the standard AT. A summary of all results is presented in Fig. 2.

---

[1]The code is publicly available at: https://github.com/EnesAltinisik/SMAAT-25/tree/main

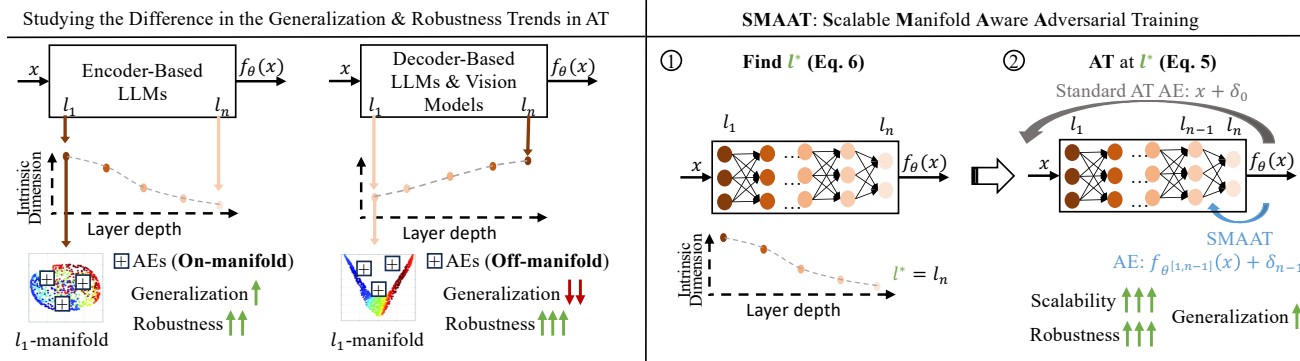

*Figure 3.* Left: In classical AT, adversarial examples (AEs) are created in the data layer. For encoder LLMs, the intrinsic dimensionality tends to be high in the initial layers and therefore the AEs created tend to be on-manifold which results in better generalization. In Vision and decoder LLMs, we observe the opposite behavior and AEs tend to be off-manifold resulting in better robustness. Right: The key idea of SMAAT is to create AEs in intermediate layers where the intrinsic dimensionality is low and AEs will tend to be off-manifold. This results in better robustness while (surprisingly) maintaining generalization. The speed-up in SMAAT is due to the fact we need shorter backprop chains to create AEs in intermediate layers.

In summary, the major contributions of our work are:

1. An explanation for the discrepancy in the robustness and generalization trends in foundation models.

2. SMAAT, a novel AT algorithm that leverages the intrinsic dimensionality across layers of foundational models to control robustness, generalization and scalability.

3. Comprehensive experiments demonstrating enhancements in robustness and scalability while keeping accuracy in classification and retrieval tasks compared to standard AT.

## 2. Related Work

**Adversarial Training (AT)** aims at robustifying a model against AEs which are imperceptibly perturbed inputs that can lead to incorrect predictions. Formally, AT seeks optimal parameters $\theta^*$ for a classifier $f_\theta(x)$ that remains robust to perturbations $\delta$ within a norm ball:

$$\min_\theta \mathbb{E}_{(x,y)\sim\mathcal{D}}\Big[\max_{\|\delta\|\leq\epsilon} \ell(f_\theta(x+\delta), y)\Big], \quad (1)$$

where $\ell$ is the loss function and $\mathcal{D} = \{(x,y)\}_{i=1}^{|\mathcal{D}|}$ represents the training data. The outer minimization is typically solved using stochastic gradient descent (SGD), while the inner maximization is addressed via projected gradient descent (PGD) (Madry et al., 2018). LAT extends AT by applying adversarial perturbations to the model's latent representations instead of its inputs (Casper et al., 2024; Sheshadri et al., 2024). Given a model with parameters $\theta = (\theta_1, \theta_2)$, it computes $h_{\theta_2} \circ g_{\theta_1}$, where $g_{\theta_1}$ is a feature extractor that maps inputs to latent representations $\ell_i = g_{\theta_1}(x_i)$, and $h_{\theta_2}$

maps these latents to predictions $\hat{y}_i = h_{\theta_2}(\ell_i)$. The standard AT objective under an $L_p$-norm constraint of $\epsilon$ is:

$$\min_\theta \sum_i \max_{\delta_i} \mathcal{L}(h_{\theta_2}(g_{\theta_1}(x_i + \delta_i)), y_i)$$

Similar to input-space AT, the inner maximization finds the worst-case perturbation $\delta_i$, while the outer minimization updates $\theta$, both solved via gradient-based methods. However, AT with $P$-step PGD is significantly more computationally expensive than standard training, as it requires $P$ forward-backward passes per update, compared to just one in standard SGD.

**Adversarial Attacks on LLMs.** Adversarial attacks on LLMs are more challenging than on vision models due to the discrete nature of text and tokenization. Early attacks on enc-LLMs used word substitutions guided by embedding similarity (Jin et al., 2020b), synonymity (Zang et al., 2020), or masked language models (Li et al., 2020), primarily to flip classifier outputs. In contrast, recent attacks on dec-LLMs focus on alignment breaking objectives by appending adversarial prefixes or suffixes. For example, AutoDAN (Liu et al., 2024) and PAIR (Wei et al., 2023) craft prompts that encourage harmful completions, while Greedy Coordinate Gradient (GCG) attack (Zou et al., 2023) appends a gradient based generated adversarial suffix to user inputs. GCG combines affirmative prompting (Wei et al., 2023; Carlini et al., 2023) with greedy and gradient-based discrete optimization (Shin et al., 2020), and is notable for its strong transferability across prompts and models.

**Manifold-Based Defenses.** The manifold conjecture stands as one of the most compelling explanations for the susceptibility of deep neural networks to AEs (Tanay & Griffin,

2016; Gilmer et al., 2018; Shamir et al., 2021). The conjecture posits that data resides on a low-dimensional manifold within a high-dimensional representation space and that the deepnet learns to approximate this manifold. Consequently, an off-manifold sample, deviating from this foundational manifold, leads to undefined behavior of the model. This conjecture has inspired a novel line of defenses against adversarial attacks on images (Samangouei et al., 2018; Meng & Chen, 2017; Song et al., 2018; Schott et al., 2019) and text (Minh & Luu, 2022). These methods approximate the data manifold and, during testing, project samples onto this manifold to either detect or correctly classify AEs. Differently, we leverage the manifold conjecture during training to improve both robustness and scalability of AT.

**Robustness and Generalization Trends in AT.** AT is recognized for enhancing model robustness in both vision and enc-LLMs (Zhang et al., 2019b; Altinisik et al., 2023b). While this improvement in robustness comes at the cost of increased generalization error in vision models (Zhang et al., 2019b), AT enhances generalization in enc-LLMs (Altinisik et al., 2023b; Gan et al., 2020). There has been recently efforts relating robustness to OFM-AEs and generalization to ONM-AEs (Stutz et al., 2019b; Xiao et al., 2025). We extend this line of research by connecting these trends to the intrinsic dimensionality of intermediate layers of a network.

**Scalable AT.** Different optimizations have been proposed to mitigate the cost of the PGD additional forward-backward passes, including (i) replacing multi-step PGD with a single-step FGSM (Shafahi et al., 2019); (ii) omitting redundant computations during PGD (Zhang et al., 2019a); (iii) combining FGSM with random initialization (Wong et al., 2020). While these approaches alleviate the PGD overhead, they also come with limitations. As a solution, FreeLB (Zhu et al., 2020) proposes accumulating model parameter gradients over multiple batches. We achieve scalability through an alternative method, *i.e.,* by leveraging the manifold conjecture to perturb intermediate layers and thus reduce the length of backward-forward PGD chains.

**ID Estimation.** It is well known that most real world data lives in a low-dimensional space relative to the ambient space where it is defined. The ID of data is the minimum number of variables necessary to characterize important properties of the data. Singular Value Decomposition (SVD) is a well known method to estimate the ID assuming the data lives on a linear manifold (Stewart, 1993). While there have been many proposals to estimate the ID in non-linear settings, we will use the relatively recent twoNN ID-estimator based on the observation that for every data point $x$, the ratio $\mu$ of distance of $x$ to it's second and first nearest neighbor follows a Pareto distribution $f(\mu|I) = I\mu^{-(I+1)}$, where $I$ is the ID. It can be estimated as $I = \frac{\log(1-F(\mu))}{\log(\mu)}$, where $F(\mu)$ is the empirical CDF (Facco et al., 2017).

## 3. Notation

Consider a deep neural network classifier $f_\theta$ with $n$ layers and parameters $\theta = \{\theta^{(i)}\}_{i=1}^n$. We define the transformation spanning layers $l_i$ to $l_j$ as $f_{\theta[i,j]} = f_{\theta(j)} \circ \cdots \circ f_{\theta(i)}$, with $\bar{f}_{\theta[i,j]}$ denoting its standardized version across the dataset $\mathcal{D}$. The input matrix $\mathcal{X} \in \mathbb{R}^{d \times |\mathcal{D}|}$ stacks all samples $\{x_i\}_{i=1}^{|\mathcal{D}|}$, while $f_{\theta[1,l]}(\mathcal{X}) \in \mathbb{R}^{d_l \times |\mathcal{D}|}$ stacks their transformed representations. The **data manifold** is defined by the high-density region of the sample distribution, distinguishing **on-manifold** (high-density) from **off-manifold** (low-density) points. Similarly, the $l^{\text{th}}$ **layer manifold** is characterized by the density of standardized representations $\{\bar{f}_{\theta[1,l]}(x_i)\}_{i=1}^{|\mathcal{D}|}$, approximated via the eigenspace of the covariance matrix $\bar{f}_{\theta[1,l]}(\mathcal{X})\bar{f}_{\theta[1,l]}(\mathcal{X})^T$. We denote its eigenvectors and eigenvalues as $(U_l, \Sigma_l)$, where $U_l \in \mathbb{R}^{d_l \times d_l}$ and $\Sigma_l \in \mathbb{R}^{d_l \times d_l}$.

The **projection error** on the top $k$ eigenvectors is:

$$e_l^k(x) = \bar{f}_{\theta[1,l]}(x) - U_l^k U_l^{k^T} \bar{f}_{\theta[1,l]}(x), \quad \forall x \in \mathcal{X}. \quad (2)$$

The **eigen-based manifold dimension** $k_l$ is the minimum $k$ required for a reconstruction error $\gamma$:

$$k_l = \min_k \left\{ k \in [1, d_l] \ \Big| \ \sum_{x \in \mathcal{D}} \|e_l^k(x)\|_2 \leq \gamma \right\}. \quad (3)$$

We classify adversarial examples (AEs) based on projection error:

- A $(\gamma, l)$-**off-manifold AE** (OFM-AE) has $e_l^k > \gamma$.

- A $(\gamma, l)$-**on-manifold AE** (ONM-AE) has $e_l^k \leq \gamma$.

Finally, we denote the **Intrinsic Dimension (ID)** of layer $l$, estimated via TwoNN, as $I_l$, and its **normalized ID** as $I_l^d = I_l/d_l$.

## 4. Exploring Layerwise ID and Its Effects

Our study is motivated by the not-well understood difference in the generalization and robustness trends between adversarially trained vision models and LLMs. Specifically, (1) robustness improvements are consistently more pronounced in vision models (Madry et al., 2018; Altinisik et al., 2023b), (2) generalization either remains stable or improves in enc-LLMs (Tsipras et al., 2019; Zhu et al., 2020), whereas it deteriorates in vision models, and (3) dec-LLMs achieve the optimal trade-off when AT is applied across multiple network layers (Casper et al., 2024; Sheshadri et al., 2024).

We demonstrate that the varying trends observed across different network architectures are closely linked to the layerwise distribution of OFM-AEs. Notably, we find that in the first layer, where conventional AT is applied, adversarial

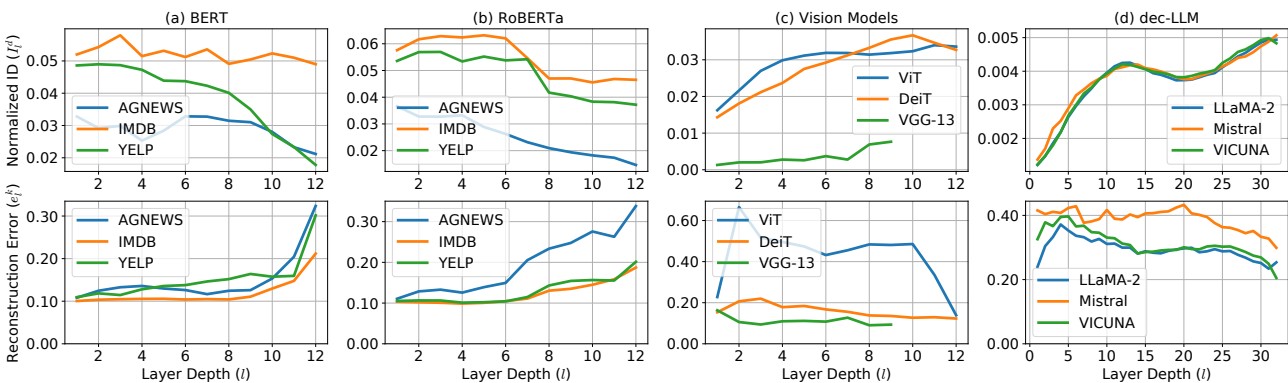

*Figure 4.* The ID (row 1) trend follows the inverse OFM-/ONM-AEs ratio (row 2) trend. The average projection error ($e_l^k$) is used as a proxy for estimating the OFM-/ONM- AEs ratio. The ID is computed using the twoNN approach. Enc-LLMs (BERT, RoBERTa) have decreasing ID and OFM-AEs proportions trends unlike vision models and dec-LLMs that have increasing ID and ONM-AEs trends.

examples for enc-LLMs tend to be more on-manifold, while vision models and dec-LLMs show a higher proportion of off-manifold examples (see Fig. 3, *left*). This explains the generalization and robustness trends as, by the manifold conjecture, higher proportions of ONM and OFM AEs lead, respectively, to better generalization and robustness (Ma et al., 2018; Stutz et al., 2019a; Alemany & Pissinou, 2020; Li et al., 2021a).

**ID & ONM/OFM Relationship** We hypothesize that lower intrinsic dimensionality ($I_l^d$) leads to a higher proportion of OFM-AEs, whereas higher $I_l^d$ yields more ONM-AEs. Formally, let $\mathcal{M} \subset \mathbb{R}^n$ be a smooth, compact, low-dimensional manifold with intrinsic dimension $I_l^d \ll n$, embedded in $\mathbb{R}^n$. Let $f : \mathbb{R}^n \to \mathbb{R}^k$ be a classifier trained on data sampled from $\mathcal{M}$. Since the classifier learns $p(y \mid x)$ without access to the true generative distribution $p(x)$, it lacks explicit knowledge of the underlying data manifold $\mathcal{M}$. Consequently, the loss gradient $\nabla_\delta \mathcal{L}$ often contains components that are orthogonal to the tangent space of $\mathcal{M}$. As a result, the perturbation $\delta$ is unlikely to remain entirely within the manifold and typically includes a non-zero component in the orthogonal direction. Moreover, when $I_l^d$ is small relative to $n$, the proportion of $\delta$ that lies off the manifold increases. Thus, adversarial examples generated via gradient-based methods are more likely to fall outside the data manifold, i.e., to be off-manifold adversarial examples.

We base this conclusion on an empirical investigation into the relationship between the ID and the proportion of ONM/OFM AEs across different layers of various deep neural network models, including two enc-LLMs (BERT and RoBERTa), three vision models (ViT (Dosovitskiy et al., 2021), DeiT (Touvron et al., 2021), VGG-13 (Simonyan & Zisserman, 2014)), and three dec-LLMs (LLaMA-2-7b-chat (Touvron et al., 2023), Vicuna (Zheng et al., 2024),

Mistral-7b-instruct (Jiang et al., 2023)) on several datasets (refer to Appendix B for details). Results are reported in Figure 4. The ID ($I_l^d$) in row 1, is computed using twoNN estimation (Facco et al., 2017) and follows the OFM-AEs trend (row 2). To assess the proportion of OFM/ONM AEs across different layers, we use the reconstruction error $e_l^k$ at layer $l$ (row 2) as a metric, *i.e.,* higher projection error corresponds to higher proportion of OFM-AEs.

**ID & Robustness/Generalization Relationship**

We extend our evaluation of the relationship between intrinsic dimensionality (ID), robustness, and generalization by applying adversarial training (AT) to different layers of dec-LLMs (LLaMA-2-7B-chat), enc-LLMs (BERT), and vision models (VGG). For the dec-LLM, we train LLaMA-2-7B-chat on the LAT dataset (Sheshadri et al., 2024) using various hyperparameter configurations (see Appendix C for details), applying LAT at every even-numbered layer. To evaluate generalization, we use MT-Bench (Zheng et al., 2024), while robustness is assessed via failure rates against the GCG attack , which generates adversarial suffixes that bypass language model safety alignment. Additional results using the PAIR (Chao et al., 2023) attack are provided in Appendix C. As shown in Fig. 5(a), applying LAT to lower layers (light blue markers) enhances robustness but reduces generalization (bottom-right), while applying it to upper layers (dark blue markers) improves generalization at the cost of robustness (top-left). This trend aligns with the intrinsic dimension (ID), which decreases in higher layers, as well as with the distribution of ONM/OFM adversarial examples. These results also explain the superiority of the LAT approach of Sheshadri et al., 2024 which applies AT to four evenly spaced layers across the network.

We conducted a similar experiment on the BERT model using the YELP dataset (see Sec. 6.1 for details). Our results

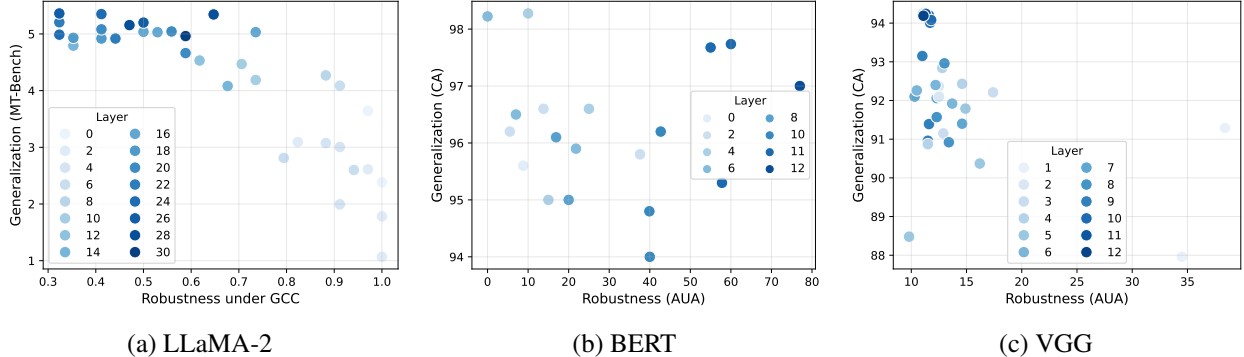

(a) LLaMA-2           (b) BERT           (c) VGG

*Figure 5.* Layer-wise effects of adversarial training on robustness and generalization across model types. Each subplot shows the impact of applying AT at different layers of (a) LLaMA-2 (dec-LLM), (b) BERT (enc-LLM), and (c) VGG (vision model). Marker colors transition from light blue (lower layers) to dark blue (higher layers). Observed trends align with changes in intrinsic dimensionality and the distribution of on- vs. off-manifold adversarial examples, as shown in Figure 4.

empirically show that, following the decreasing ID trend, applying AT at higher layers leads to increased robustness as the layer index $i$ increases. Figure 5(b) presents the results of AT on BERT for the YELP dataset. As expected, robustness (measured under the TextFooler attack) improves with higher layer indices while generalization remains unaffected.

We also extend this analysis to vision models. We selected VGG-13 and trained it on CIFAR-10 (Krizhevsky, 2009) using AT applied to every ReLU layer over 20 epochs. We varied the attack strength ($\epsilon = 0.031$–$0.2$) and learning rate ($lr = 0.01$–$0.001$), evaluating robustness via Robust-Bench (). As shown in Fig. 5(c), adversarial training on lower layers (light blue) improves robustness but reduces generalization, whereas training on higher layers (dark blue) yields the opposite pattern. These results suggest that vision models resemble dec-LLMs in terms of their generalization–robustness tradeoff. Finally, our findings are consistent with those shown in Fig. 4, where a higher ratio of OFM-AEs in deeper layers correlates with improved robustness in both enc-LLMs and vision models.

Beyond providing an explanation for the difference in robustness and generalization trends in vision models, enc- and dec-LLMs, results of Fig. 4 shed light on an interesting trend for the ID across layers. While the ID is increasing in vision and dec-LLMs, it is monotonically decreasing in enc-LLMs (Bert and RoBERTa). This motivated the design of a new algorithm that controls the gains in robustness and generalization via controlling the proportions of ONM- and OFM-AEs. This can be achieved by generating AEs in the intermediate layers of the deepnet based on their IDs. To this end, we propose SMAAT (See Fig. 3, *right*), an AT algorithm that aims at achieving high robustness by perturbing the layer $l^*$ with the lowest ID to generate a high proportion of OFM-AEs. A side effect of perturbing an intermediate layer as opposed

to the input one is significantly improving scalability by a factor $\mathcal{O}\Big( P(n - l^*)\Big( \max\Big( \{d_{l_i} | l_i \in [1, l^* - 1]\} \Big) - d_{l^*} \Big) \Big)$ as it results in shorter $P$-step PGD chains.

## 5. SMAAT: Scalable Manifold Aware Adversarial Training

We propose SMAAT an AT algorithm that aims at improving the robustness and scalability of standard AT by generating AEs in the layer leading to the highest proportion of OFM instead of from the input layer as classically. Specifically, given a pretrained model $f_\theta$, SMAAT intentionally generates a higher proportion of OFM-AEs to enhance robustness. Formally, SMAAT solves the augmented AT objective:

$$\min_\theta \quad \mathbb{E}_{(x,y)\sim\mathcal{D}} \left[ \max_{\|\delta\|\leq\epsilon} \ell(f_\theta(x + \delta), y) \right] \tag{4}$$
$$\text{s.t.} \quad (x + \delta) \text{ is } (\gamma, 1)\text{-OFM}.$$

While the manifold conjecture refers to input space AEs, we relax it to encompass intermediate layers as well, *i.e.,* we chose the perturbation $\delta$ that results in an OFM-AE at any layer across the deepnet. Intuitively, AEs that are off- or on- the transformed data manifold in any layer also affect robustness. The relaxed objective becomes:

$$\min_\theta \quad \mathbb{E}_{(x,y)\sim\mathcal{D}} \left[ \max_{\|\delta\|\leq\epsilon} \ell(f_\theta(x + \delta), y) \right] \tag{5}$$
$$\text{s.t.} \quad \exists l \in [1, L] : (x + \delta) \text{ is } (\gamma, l)\text{-OFM}.$$

Note that solving the above objective using the method of Lagrangian Multipliers is possible, but it would require an approximation of the manifold to characterize OFM-AEs. Such approximations are either computationally expensive (*e.g.,* GAN (Xiao et al., 2018)) or overly simplistic (*e.g.,* eigenbasis (Xiao et al., 2025)). SMAAT uses an alternative approach to find OFM-AEs by perturbing the layer with the

*Table 1.* Robustifying sentiment classifiers on AGNEWS, IMDB, and YELP datasets. We report the clean accuracy (CA) and the robust accuracy under attack (AUA) with PWWS (PW), TextFooler (TF), and Bert-Attack (BA), along with the average robust accuracy (AR) across these three attacks. The best performance is **bold**ed. Across all data sets, SMAAT achieves high robustness (AR) while maintaining high generalization (CA).

| Model | Defense | AGNEWS | | | | | IMDB | | | | | YELP | | | | |
|---|---|---|---|---|---|---|---|---|---|---|---|---|---|---|---|---|
| | | CA | AUA | | | | CA | AUA | | | | CA | AUA | | | |
| | | | PW | TF | BA | AR | | PW | TF | BA | AR | | PW | TF | BA | AR |
| BERT | Standard | 94.5 | 36.9 | 28.1 | 37.5 | 34.2 | 92.2 | 15.0 | 5.8 | 5.4 | 8.7 | **97.0** | 12.2 | 6.5 | 5.3 | 8.0 |
| | ASCC | 91.6 | 32.8 | 31.4 | 32.1 | 32.1 | 88.5 | 15.1 | 12.4 | 11.2 | 12.9 | 91.5 | 19.4 | 15.7 | 12.2 | 15.8 |
| | FreeLB++ | **95.1** | 47.9 | 51.5 | 41.8 | 47.1 | **93.2** | 12.5 | 45.3 | 39.9 | 32.6 | 95.6 | 19.3 | 8.8 | 3.7 | 10.6 |
| | SAFER | 94.4 | 39.3 | 35.5 | 42.3 | 39.0 | 92.3 | 41.4 | 39.1 | 30.7 | 37.1 | 95.4 | 29.8 | 25.8 | 23.7 | 26.4 |
| | TMD | 94.3 | 70.0 | 50.0 | 55.2 | 58.4 | 92.2 | 38.7 | 44.2 | 33.7 | 38.9 | 95.2 | 36.8 | 40.9 | 28.6 | 35.4 |
| | RSMI | 92.7 | **76.1** | 63.2 | NA[2] | NA | 92.2 | 58.7 | 56.4 | NA[2] | NA | 95.4 | 45.3 | 52.3 | NA[2] | NA |
| | **SMAAT (Ours)** | 94.6 | 73.5 | **72.2** | **74.7** | 73.5 | 92.2 | **63.6** | 77.9 | 60.8 | 67.4 | **97.0** | **77.1** | **77.9** | **72.8** | 75.9 |
| RoBERTa | Standard | 94.7 | 30.6 | 23.9 | 37.1 | 30.5 | 94.0 | 8.7 | 2.1 | 0.6 | 3.8 | 97.9 | 23.1 | 14.9 | 9.0 | 15.7 |
| | ASCC | 92.6 | 48.1 | 41.0 | 49.1 | 46.1 | 92.6 | 23.1 | 13.5 | 11.8 | 16.1 | 95.4 | 15.0 | 8.6 | 4.5 | 9.4 |
| | FreeLB++ | **95.6** | 61.0 | 49.8 | 56.6 | 55.8 | **94.3** | 33.6 | 14.6 | 6.1 | 18.1 | 97.0 | 38.6 | 46.0 | 35.2 | 39.9 |
| | SAFER | 94.6 | 68.9 | 49.3 | 46.1 | 54.8 | 93.9 | 52.8 | 47.1 | 40.6 | 46.8 | 96.6 | 65.6 | 67.9 | 48.3 | 60.6 |
| | TMD | 95.0 | 68.3 | 54.0 | 56.7 | 59.7 | 93.3 | 60.5 | 66.8 | 51.6 | 59.6 | 96.6 | 68.9 | 70.9 | 51.0 | 63.6 |
| | RSMI | 94.3 | **81.9** | 74.1 | NA[2] | NA | 93.0 | 76.2 | 73.4 | NA[2] | NA | 96.3 | 68.9 | 65.9 | NA[2] | NA |
| | **SMAAT (Ours)** | 94.6 | 75.6 | **75.1** | **79.9** | 76.9 | 93.5 | **77.1** | **78.5** | **63.2** | 72.9 | **98.0** | **85.4** | **86.4** | **76.0** | 82.6 |

lowest ID without the need for manifold approximations. Particularly, SMAAT applies AT at $l^*$-th layer where the layer with more OFM-AEs composition (lowest ID):

$$\min_\theta \mathbb{E}_{(x,y)\sim\mathcal{D}} \left[ \max_{\|\delta_{l^*}\|\leq\epsilon_{l^*}} \ell\Big(f_{\theta^{[l^*,n]}}\Big(f_{\theta^{[1,l^*]}}(x)+\delta_{l^*}\Big),y\Big) \right]. \tag{6}$$

We additionally choose $l^*$ to correspond to the layer with the highest index as this would lead to better scalability (shorter PGD chains). as well as to potentially the highest proportion of off-manifold AEs at any intermediate layer, *i.e.,*

$$l^* = \max_l \left\{ l \in [1,n] \Big| I_l^* \leq I_i \forall i < l \right\}. \tag{7}$$

**Complexity.** Computing the layers IDs and searching for the optimal layer $l^*$ to perturb are done once per model and per task. Thus, they incur marginal overhead. For the AT part, when a $P$-step PGD attack is used, this results in $P$ forward-backward passes with length $(n-l^*+1)$ instead of $n$. The run-time of every forward/backward pass depends on the layer dimensionality, *i.e.,* $O(d_{l-1} \times d_l)$ for the $l^{\text{th}}$ layer. Overall, the complexity of one SMAAT forward-backward is $\mathcal{O}((n-l^*+1)\max_{l\in[l^*,n]}(d_l)^2))$. As a result, SMAAT is more efficient than standard AT by a factor of $\mathcal{O}(P \times l^* \cdot (\max_{i\in[1,n]}(d_i)^2 - \max_{j\in[l^*,n]}(d_j)^2))$. In the case Enc-LLMs (*e.g.,* BERT and RoBERTa), $l^*$ is equal to $n$. The total run-time can be simplified to $\mathcal{O}(P \cdot (d_n)^2)$ where $d_n$ is the number of classes. Typical enc-LLMs tasks consist of less than five classes (Wang et al., 2019a) which makes the factor

of classes small enough to be negligible. Hence, SMAAT enhances the efficiency of the AEs generation process by a factor of $l \cdot O(\max(d_l))$. This improvement practically eliminates the cost of the AE generation process.

# 6. Experiments

We assess robustbness, generalization and scalability of three types of models trained with SMAAT: (i) classifiers; (ii) retrievers in RAG settings and (iii) safeguarding models employed for moderating content produced by generative models. We consider four different attacks: (i) word substitution (Ren et al., 2019; Jin et al., 2020a; Li et al., 2021b), (ii) adaptive (Tramèr et al., 2018), (iii) Greedy Coordinate Gradient (GCG) (Zou et al., 2023), and (iv) corpus poisoning (Zhong et al., 2023) attacks. We include a run-time analysis to demonstrate scalability of SMAAT.

## 6.1. Sentiment and Topic Classifiers.

We adversarially trained two **base models**, BERT-base-cased and RoBERTa-base-cased on the **tasks** of sentiment classification (IMDB (Maas et al., 2011) and YELP (Zhang et al., 2015)) and topic classification (AGNEWS (Zhang et al., 2015)) under **three input space attacks** including PWWS (Ren et al., 2019) (synonym based), TextFooler (Jin et al., 2020a) (neighbor based), and BERT-Attack (Li et al., 2021b) (masked-LM based). Attacks are conducted

*Table 2.* Run-time results on IMDB dataset. Mean and standard deviation are computed over ten runs. Compared to FreeLB++, training time of SMAAT is lower by a factor of nearly three.

| | Standard | ASCC | FreeLB++ | SAFER | RSMI | **SMAAT** |
|---|---|---|---|---|---|---|
| Training (min/epoch) | 5.1 ±0.1 | 25.7 ±0.3 | 15.6 ±0.5 | 8.2 ±0.6 | 15.4 ±0.3 | 5.2 ±0.2 |
| Inference (msec/sample) | 2.4 ±0.1 | 41.4 ±0.2 | 2.4 ±0.0 | 2.4 ±0.0 | 5.6 ±0.4 | 2.4 ±0.1 |

*Table 3.* Robustifying dec-LLMs with encoder-based (BERT etc.) safety filters on AdvBench and HH_RLHF datasets. We report clean accuracy (CA), and robust accuracy under attack (AUA) which indicates the percentage of harmful prompts augmented with adversarial suffices accurately classified as harmful. **Notice that SMAAT achieves very high robustness while other models sometimes completely fail**.

| Dataset | Model | BERT | | RoBERTa | | DistilBERT | |
|---|---|---|---|---|---|---|---|
| | | CA | AUA | CA | AUA | CA | AUA |
| AdvBench | Standard | **100** | 0 | **100** | 0 | 99.6 | 0 |
| | FreeLB++ | 99.6 | 0 | **100** | 0 | **99.6** | 0 |
| | **SMAAT** | **100** | **100** | **100** | **99.2** | 99.2 | **97.5** |
| HH-RLHF | Standard | 95.4 | 26.4 | 94.8 | 0.3 | **98.7** | 6.5 |
| | FreeLB++ | **98.6** | 48.5 | **98.6** | 34.0 | 98.5 | 34.1 |
| | **SMAAT** | 95.4 | **51.6** | 94.8 | **96.8** | 98.5 | **39.5** |

using the TextAttack framework (Morris et al., 2020) and following the settings introduced by (Li et al., 2021b). We compare SMAAT to standard (non-adversarial) training, and six **baselines** from three families of defenses: (1) Input space AT (ASCC (Dong et al., 2021)), (2) Embedding space AT (FreeLB++ (Li et al., 2021b)), and (3) Certified defenses (SAFER (Ye et al., 2020), TMD (Minh & Luu, 2022), RSMI (Minh & Luu, 2022)). FreeLB++ focuses on scalability and robustness by minimizing the number of PGD steps and applying AT in the initial layer. TMD leverages manifold features by projecting samples back to the manifold in the last layer. **Implementation details:** In both base models, $l^* = n$, *i.e.,* we generate AEs in the last layer. Specifically, we perturb the [CLS] embeddings before the classifier layer by freezing all layers before $l^*$. Further details are presented in Appendix D.

**Robustness and generalization:** In Table 1, we report results on robustness and generalization. On average, SMAAT demonstrates superior robustness over all datasets, with an improvement of 8.6%, 15.7%, and 28.8% over the best score for the BERT model, and 6.0%, 5.8%, and 19.0% for the RoBERTa model on the AGNEWS, IMDB, and YELP datasets, respectively. Note that FreeLB++, which perturbs the first layer, showed the best generalization in four out of six cases, as it produces mostly on-manifold examples (the first layer has a high dimension, as illustrated in Fig. 4) which is in line with the manifold conjecture. SMAAT maintains generalization in five out of six cases and only shows

0.5 drop in performance in the case of RoBERTa with the IMDB dataset. Besides, SMAAT consistently outperforms TMD, even though it is a manifold-based method differently from SMAAT, TMD estimates the manifold and projects the input samples onto it before classification witch could lead to inaccuracies in the manifold estimation. SMAAT on the other hand is more robust to such inaccuracies as it only leverages the ID for selecting the layer to perturb. RSMI shows strong robustness against PWWS, as its masked inference mechanism inherently mimics synonym-based defenses without needing an explicit synonym set (Minh & Luu, 2022).

**Runtime efficiency:** In Table 2, we provide details on the training time per epoch and inference time per instance for the IMDB dataset using BERT (RoBERTa has the same architecture). SMAAT has comparable efficiency to standard training and is on average 3 times faster than standard AT during training. This is attributed to shorter backpropagation chains as AT in SMAAT is performed in the last layer ($l^* = n$). Note that FreeLB++, which injects noise in the first layer, remains inefficient even when PGD is replaced with FGSM (Zhu et al., 2020; Li et al., 2021b), as a full-depth backpropagation chain is still required. Certified defense baselines (SAFER, RSMI) and input space attack (ASCC) are also time-consuming as they either require mapping samples into the manifold or performing an extensive search over word substitutes. Additional evaluation of SMAAT on **language understanding benchmarks**, including GLUE and advGLUE are reported in Appendix E.

## 6.2. Safety Filters

The broad attack surface of LLMs has compelled model owners to implement solutions that extend beyond the safety alignment of a model. Notably, this includes content moderation filters that verify the harmlessness of a model's inputs and outputs (Kumar et al., 2023; Cao et al., 2024; Fatehkia et al., 2025). These filters are essentially text classifiers, often based on lightweight enc-LLMs to minimize overhead and preserve the high responsiveness of dec-LLMs (Kumar et al., 2023). The robustness of safety filters is also a concern, as they too can be vulnerable to attacks. **Base models:** We evaluate SMAAT's effectiveness on BERT, RoBERTa,

---

[2]RSMI takes about 2k times longer than TextFooler to generate a AE with BERT-Attack, making it unfeasible to test.

*Table 4.* Robustifying retriever models of RAG on the Natural Questions dataset. We report Recall@10 (R@10) and Recall@100 (R@100) on the clean corpus for generalization. Robust recall (RR) is measured by how many samples are selected without adversarial passages in the top-k passages, with R@10 and R@100 corresponding to the top-10 and top-100 passages, respectively. During attacks, 10 and 50 adversarial passages are created, denoted as (N=10) and (N=50), respectively. SMAAT makes RAG substantially more robust against selecting adversarial passages.

| Model | Generalization | | RR (N=10) | | RR (N=50) | |
|---|---|---|---|---|---|---|
| | R@10 | R@100 | R@10 | R@100 | R@10 | R@100 |
| Standard | 36.4 | 79.7 | 46.0 | 22.2 | 26.1 | 7.2 |
| FreeLB++ | **39.6** | **81.8** | 75.2 | 60.7 | 51.7 | 32.2 |
| **SMAAT** | 34.9 | 79.5 | **99.5** | **97.1** | **85.7** | **73.5** |

and DistilBERT when used as safety filters. **Datasets:** The AdvBench dataset (Zou et al., 2023) and the Helpfulness-Harmfulness dataset (HH-RLHF) (Bai et al., 2022). **Adversarial attack:**, we use GCG attack. **Metrics:** we assess generalization accuracy (ACC) for safe and harmful prompts, and robust accuracy under attack (AUA) by measuring the detection rate of harmful prompts augmented with adversarial suffixes. Further details are presented in Appendix F. **Robustness and generalization:** Results in Table 3 show that SMAAT significantly enhances the robustness of safety filters against the GCG attack across all models, while maintaining generalization compared to standard training. Also, FreeLB++ improves generalization as it applies AT in the first layer.

### 6.3. Retriever Models of RAG

RAG combines a retriever model, which identifies relevant passages from a large corpus, with a generator model that constructs answers based on the retrieved information. **Base model:** We use the Contriever model (Izacard et al., 2022), fine-tuned on the Natural Questions (NQ) (Kwiatkowski et al., 2019) dataset, as our retriever. **Baselines:** We evaluate the robustness of retriever models within the RAG framework under standard, FreeLB++ and SMAAT. **Adversarial attacks:**, we use poisoning attacks (Zhong et al., 2023), which manipulate the retrieval corpus by generating adversarial passages. As **metrics**, we use robust recall (RR) measured by how many samples are selected without adversarial passages in the top-k passages, with R@10 and R@100 corresponding to the top-10 and top-100 passages, respectively. **Robustness and generalization:** Table 4 shows that the standard model offers limited robustness against the attack, with a significant number of adversarial passages being retrieved. FreeLB++ shows some improvement, reducing the number of adversarial passages retrieved. However, SMAAT significantly enhances robustness, demonstrating a dramatic reduction in the retrieval of

adversarial passages compared to the others. In terms of generalization, FreeLB++ yields the best results as it applies AT in the first layer (resulting in more ONM-AEs), while both SMAAT and standard training exhibit similar performance.

## 7. Conclusion

In this paper, we explain the varying effects of AT across architectures using the manifold conjecture, showing that ID influences robustness and generalization. Vision and decoder-based models favor off-manifold AEs in early layers, enhancing robustness but harming generalization, while encoder-based LLMs favor on-manifold AEs, preserving generalization with limited robustness gains. Leveraging this insight, we introduce SMAAT, which improves AT scalability by perturbing the layer with the lowest ID. This reduces training overhead by 25–33% while boosting robustness across sentiment classification, safety filtering, and retrieval tasks, maintaining generalization. We plan to explore joint optimization of generalization and robustness by controlling the on/off-manifold AE ratio through perturbations in intermediate layers.

## Acknowledgement

This work is partial supported by the Qatar National Research Fund (QNRF) grant NPRP11C-1229-170007.

## Impact Statement

This research aims to deepen the understanding of how AT methods influence generalization and robustness, leading to more effective and practical adversarial training techniques. As machine learning models become increasingly integrated into real-world applications, it is crucial to develop frameworks that mitigate misuse while ensuring ethical compliance. Building on prior work in adversarial training, our approach enhances the robustness of encoder-based models while significantly improving computational efficiency.

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

# A. Supplementary Material

In this paper, we propose SMAAT, an AT algorithm that not only optimizes for better robustness and generalization but also enhances scalability. Specifically, SMAAT leverages the manifold conjecture, which posits that OFM-AEs lead to better robustness while ONM-AEs enhance generalization. To achieve this, SMAAT perturbs the layer with the lowest intrinsic dimension (ID). Intuitively, perturbing this layer would yield the highest proportion of OFM-AEs across layers. Formally, SMAAT solves the following program:

$$\min_{\theta} \mathbb{E}_{(x,y)\sim\mathcal{D}} \Big[ \max_{\|\delta_{l^*}\|\leq\epsilon_{l^*}} \ell\Big( f_{\theta^{[l^*,n]}}\Big( f_{\theta^{[1,l^*]}}(x) + \delta_{l^*}\Big), y\Big)\Big].$$

We show that $l^*$ corresponds to the last layer in enc-LLMs and the first layer in dec-LLMs. This explains the difference in robustness and generalization trends between vision models and enc-LLMs (Stutz et al., 2019a; Shafahi et al., 2019; Zhu et al., 2020). Specifically, in vision models, improvements in robustness are often accompanied by a decrease in generalization. In contrast, in enc-LLMs, improvements in generalization are achieved with lesser gains in robustness. The algorithm for SMAAT is provided in Alg. 1.

---

**Algorithm 1** SMAAT

---

1: **Input:** $\mathcal{D} = \{\mathcal{X}, \mathcal{Y}\}$: input data, $f_\theta$: a deepnet model, $\epsilon$: attack strength, $E$: the number of epochs, $\alpha$: PGD learning rate, and $\Pi$: the projection operator into the $\epsilon$-ball.
2: **Output** $f_\theta$: a deepnet model
3:    *% Determine the ID behaviour of the model with twoNN*
4:    *% Identify the optimal layer $l^*$ to perturb (Eq. (7))*
5: **for** $e = 1, .., E$ **do**
6:    **for** $(x_i, y_i) \in \mathcal{D}$ **do**
7:       $\delta_{l^*} \sim \mathcal{N}(0, \sigma^2 I)$ *% sample an initial perturbation*
8:       *% store forward pass for scalability*
9:       $mid\_rep \leftarrow f_{\theta^{[0,l^*]}}(x_i)$
10:      **for** $s = 1, .., S$ **do**
11:        $loss \leftarrow \ell\left(f_{\theta^{[l^*,n]}}(mid\_rep + \delta_{l^*}), y_i)\right)$
12:        $\delta_{l^*} \leftarrow \Pi_{\epsilon_{l^*}}\left(\delta_{l^*} + \alpha \cdot sign\left(\nabla_{\delta_{l^*}}(loss)\right)\right)$
13:      **end for**
14:      *% update the model parameters*
15:      $loss \leftarrow \ell\left(f_{\theta^{[l^*,n]}}(mid\_rep + \delta_{l^*}), y_i)\right)$
16:      $\theta^{[l^*,n]} \leftarrow \theta^{[l^*,n]} - lr\nabla_{\theta^{[l^*,n]}}(loss)$
17:    **end for**
18: **end for**

---

SMAAT has the advantage of being significantly more runtime-efficient than classical adversarial training. The gain is of the order of $\mathcal{O}\Big( P(n - l^*)\Big( \max\Big( \{d_{l_i} | l_i \in [1, l^* - 1]\}\Big) - d_{l^*}\Big)\Big)$, for an $n$-layered deepnet under a $P$-step PGD attack. $d_l$ is the dimensionality of the layer $l$.

Empirical results demonstrate that SMAAT leads to better robustness while maintaining comparable accuracy to standard training. We achieve compelling results on robustifying (1) sentiment classifiers, (2) safety filters in decoder-based models, and (3) retriever models in the Retrieval Augmented Generation (RAG) setup.

The remaining of the supplementary material is organized as follows:

- Appendix B: Intrinsic Dimension Estimation

- Appendix C: Additional Results of ID & Robustness/Generalization Relationship

- Appendix D: Additional Results of Robustifying Sentiment Classifiers

## B. Intrinsic Dimension Estimation

In the following, we provide additional details on the experimental setup on OFM-/ONM- AEs ratio calculation and the ID estimation in Fig. 4.

**Experimental setup.** We conduct experiments on two enc-LLMs (BERT and RoBERTa), across three different datasets (AGNEWS, IMDB, YELP). Additionally, we evaluated three vision models (ViT (Dosovitskiy et al., 2021), DeiT (Touvron et al., 2021), VGG-13 (Simonyan & Zisserman, 2014)) on the CIFAR-10 (Krizhevsky, 2009) dataset and three LLMs (LLaMA-2-7b (Touvron et al., 2023), Vicuna (Zheng et al., 2024), Mistral-7b (Jiang et al., 2023)) on the HelpSteer dataset (Wang et al., 2023). We utilize the train split of the datasets to estimate the ID and eigenspace of the layers. We extract token representations from various layers of the models similar to the implementation in NeuroX (Dalvi et al., 2023). AEs are generated using TextFooler for LMs and PGD for vision models on the respective test splits of the datasets. For LLMs, AEs are generated using the Greedy Coordinate Gradient (GCG) attack (Zou et al., 2023) on the harmful prompt datasets released in the same paper. In our calculation, we use CLS embedding for BERT, RoBERTa, ViT, and DeiT models; last token embedding for LLMs; and conventional layer output for VGG-13.

**Discussion of results.** We present the average projection error, $e_l^k$, of each layer on the first row of Fig. 4. Results in Fig. 4(a,b) indicate that for LMs, the average $e_l^k$ monotonically increases, suggesting that examples become more off-manifold at the latest layers, consistent with our hypothesis. Conversely, for vision models and dec-LLMs in Fig. 4(c,d), we observe the opposite characteristic with a lower average $e_l^k$ at the latest layers as expected. The only unexpected behavior observed in vision models and dec-LLMs, except for VGG-13 and Mistral, is that they exhibit the highest off-manifold ratio ($e_l^k$) in the initial layers rather than the first layers. For transformer models, the CLS/last token embedding starts with a specific value and gradually evolves to represent the sentence. This means it takes time for the effect of AEs to manifest in the CLS/last token embedding. In contrast, since the VGG model's representation is directly calculated from the input, the highest off-manifold ratio is obtained at the first layer.

We hypothesize that this phenomenon can be explained by the ID of the layers. Specifically, if $I_l \ll d_l$, AEs tend to be off-manifold. To validate this hypothesis for the same models and dataset, we measure $I_l$ using the twoNN method and normalize it with $d_l$. The normalized $I_l$ can be seen in the second row of Fig. 4. In line with our hypothesis, while it decreases for the LMs (Fig. 4(a,b)), it increases for vision models and dec-LLMs(Fig. 4(c,d)).

## C. Additional Results of ID & Robustness/Generalization Relationship

We further examine the relationship between intrinsic dimensionality (ID), robustness, and generalization by applying Latent Adversarial Training (LAT) to different layers of the LLaMA-2-7B model. Unlike untargeted LAT, which disrupts model behavior, we employ targeted LAT to induce specific adversarial responses, following Casper et al. (2024). This is achieved by perturbing the residual stream with L2-norm-bounded noise using PGD.

To stabilize training and mitigate unintended effects, we interleave LAT with supervised fine-tuning on the UltraChat dataset (Ding et al., 2023). We evaluate generalization using MT-Bench (Zheng et al., 2024) and robustness using GCG-based and PAIR attacks from HarmBench (Mazeika et al., 2024). The model is trained with a learning rate of $2e-4$, applying LAT at every even-numbered layer with norm bounds ranging from 1 to 5.

The results are presented in Figure 6. Attacking lower layers (light blue markers) improves robustness but reduces generalization, as indicated by points concentrated in the bottom-right region. In contrast, attacking upper layers (dark blue markers) enhances generalization at the cost of robustness, shifting results toward the top-left. Additionally, while model robustness drops to as low as 30% under the GCG attack, it remains consistently above 75% across all configurations under the PAIR attack.

## D. Additional Results of Robustifying Sentiment Classifiers

In the following, we provide more details about the experiments on robustifying sentiment classifiers.

**Datasets.** We evaluate `SMAAT`on three datasets: AG-News Corpus (AGNEWS) (Zhang et al., 2015), Internet Movie

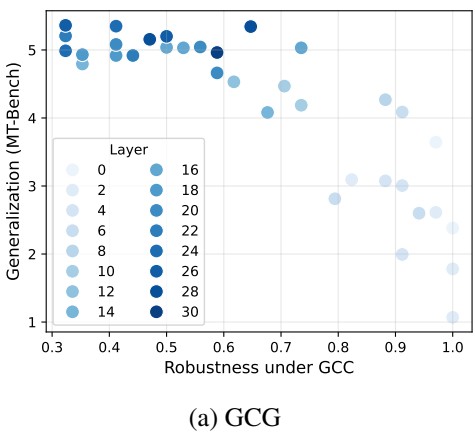

(a) GCG

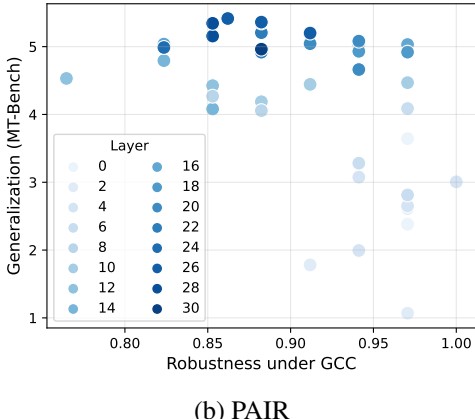

(b) PAIR

*Figure 6.* Layer-wise effects of adversarial training on generalization and robustness against GCG and PAIR attacks on the LLaMA-2 model. Marker colors represent layer depth, transitioning from light blue (lower layers) to dark blue (higher layers). The results show that robustness drops to as low as 30% under the GCG attack, while remaining above 75% under the PAIR attack across all configurations.

Database (IMDB) (Maas et al., 2011), and Yelp Review Polarity (YELP) (Zhang et al., 2015). The AGNEWS dataset contains over 120000 samples, each belonging to one of the four labels: World, Sports, Business, Sci/Tech. The IMDB dataset contains 50000 data samples of movie reviews with binary labels for negative and positive sentiments. The YELP dataset contains nearly 600000 samples of highly polar Yelp reviews with binary labels. However, due to limitations in computing resources, we only use a subset of 63000 samples of the YELP dataset. In addition, we randomly sample 10% of the training set for validation in all datasets. For testing, we use a subset of 1000 test samples from each dataset, following previous work practices. The AGNEWS dataset contains over 120k samples, categorized into four classes: World, Sports, Business, and Sci/Tech. The IMDB dataset consists of 50k movie reviews, each labeled with binary sentiments (positive or negative).

**Base model.** We employed the BERT$_{\text{base-cased}}$ (Devlin et al., 2019) and RoBERTa$_{\text{base-cased}}$ (Liu et al., 2019) models in our experiments. To conduct the evaluations, we utilize the fine-tuned models provided by *TextAttack* from HuggingFace for all datasets, except for the RoBERTa base model fine-tuned on YELP dataset. For the YELP dataset, we created a fine-tuned RoBERTa model for 2 epochs with a learning rate of $1e-05$ and a batch size of 32.

**Adversarial Attacks.** which include the following constraints: (1) The maximum percentage of modified words is set to 0.3 for AGNEWS, 0.1 for IMDB and YELP datasets, respectively. (2) For word replacement, a maximum of 50 candidates are considered for each word. (3) The semantic similarity, measured using the Universal Sentence Encoder (Cer et al., 2018), between the original input and the generated adversarial example must exceed 0.84. PWWS uses word synonyms, TextFooler applies nearest neighbor search in counter-fitting embeddings (Mrkšić et al., 2016), and BERT-Attack utilizes BERT masked language model to generate candidate words.

**Baselines.** For input space adversarial training, we consider Adversarial Sparse Convex Combination (ASCC) (Dong et al., 2021) which model the perturbation space as the convex hull of word synonyms. ASCC incorporates an entropy-based sparsity regularizer to capture word substitution geometry more effectively. In our investigation of embedding space adversarial training which recognized as the most impactful technique for enhancing generalization (Li et al., 2021b), we conduct a thorough analysis of FreeLB++ (Li et al., 2021b) (employs gradient-guided perturbations centered around the most susceptible data points). For certified defenses, we evaluate SAFER (Ye et al., 2020), TMD (Minh & Luu, 2022), and RSMI (Minh & Luu, 2022). SAFER constructs a set of randomized inputs by performing random synonym substitutions and using the statistical properties of predicted labels to certify robustness. TMD employs infoGAN (Chen et al., 2016) to project adversarial examples to the data manifold in the last layer to address the manifold issue. RMSI combines these ideas by applying importance-based masking to tokens and leveraging randomized smoothing in each layer.

**Implementation details.** To train the last layer of $f_\theta$ with adversarial samples, we create adversarial samples using 5-step PGD attacks. During training, we use epsilon values of 0.1, 0.1, and 0.8 for the YELP, AGNEWS, and IMDB datasets, respectively, for the BERT models. For the RoBERTa models, we employ epsilon values of 0.1, 0.6, and 0.03. All models are trained 10 epochs with a learning rate of 0.1. In our evaluation, we use a V100 GPU with 32 GB memory and 64 CPUs.

*Table 5.* Average accuracy from the standard training, FreeLB++ and SMAAT on GLUE and AdvGLUE datasets. Results clearly demonstrate that SMAAT enhances model generalization (GLUE results) and robustness (AdvGLUE results).

| Dataset | BERT | | | RoBERTa | | |
|---|---|---|---|---|---|---|
| | Standard | FreeLB++ | SMAAT (Ours) | Standard | FreeLB++ | SMAAT (Ours) |
| GLUE | 85.9 | 86.3 | 86.3 | 89.3 | 89.6 | 89.7 |
| AdvGLUE | 39.5 | 42.5 | 45.1 | 27.5 | 37.1 | 39.6 |

*Table 6.* Hyperparameters in experiments on robustifying the safety filters on decoding LLMs experiments. Learning rate (lr) and $\epsilon$ values for SMAAT and FreeLB++ methods.

| Dataset | FreeLB++ | | | | | | SMAAT | | | | | |
|---|---|---|---|---|---|---|---|---|---|---|---|---|
| | BERT | | RoBERTa | | DistilBERT | | BERT | | RoBERTa | | DistilBERT | |
| | lr | $\epsilon$ | $lr$ | $\epsilon$ | $lr$ | $\epsilon$ | lr | $\epsilon$ | $lr$ | $\epsilon$ | $lr$ | $\epsilon$ |
| AdvBench | 5e-3 | 0.05 | 5e-3 | 0.05 | 1e-2 | 0.01 | 1e-4 | 1e-3 | 1e-4 | 1e-3 | 1e-4 | 1e-4 |
| HH-RLHF | 5e-3 | 0.5 | 53-3 | 0.05 | 5e-4 | 0.05 | 1e-4 | 5e-4 | 1e-4 | 1e-3 | 1e-4 | 5e-4 |

# E. Additional Results of Language Understanding Benchmarks

To comprehensively evaluate SMAAT's performance against a broader spectrum of textual adversarial attacks, we employ the GLUE and AdvGLUE benchmarks. The GLUE benchmark (Wang et al., 2019a) is a comprehensive evaluation suite featuring seven diverse NLP tasks to assess model performance. The AdvGLUE benchmark (Wang et al., 2021) is an extension of GLUE, incorporating 17 distinct textual adversarial attacks, covering word-level transformations, sentence-level manipulations, and human-written AEs. This extension ensures a thorough evaluation encompassing various adversarial linguistic phenomena. For our assessment, we employ the evaluation sets of four datasets across three different tasks: Sentiment Analysis (SST-2), Duplicate Question Detection (QQP), and Natural Language Inference (QNLI, RTE).

In our evaluation, we compare SMAAT against standard BERT and RoBERTa models[3], as well as their FreeLB++ incorporated versions. In the case of SMAAT, we conducted a grid search for the learning rate, ranging from 0.1 to 0.001, and the $\epsilon$ value, ranging from 0.8 to 0.01, using 3-PGD steps As shown in Table 5, SMAAT demonstrates a robustness improvement of 5.6% and 2.6% for BERT, and 12.1% and 2.5% for RoBERTa, compared to the standard and FreeLB++ models, respectively, while maintaining similar generalization performance.

# F. Additional Results of Robustifying Safety Filters In Decoder Based LLMs

The GCG attack is deliberately designed to bypass the safety alignment of LLMs by generating a response to a potentially harmful prompt through appending an adversarial suffix to the user's input. This strategy leverages previous methodologies by using (1) an affirmative response tactic (Wei et al., 2023; Carlini et al., 2023) to direct the model's output towards the attacker's intended outcome (used for loss calculation) and (2) a mix of greedy and gradient-based discrete optimization (Shin et al., 2020) to pinpoint the most susceptible tokens. A key attribute of the GCG attack is its transferability, demonstrating that adversarial suffix designed for a specific prompt on one model can successfully affect a broad range of other models.

A suggested defense against the GCG attack involves incorporating a lightweight binary classifier model designed to identify harmful prompts (Kumar et al., 2023). It is important to note, however, that these classifiers can still be vulnerable to such attacks. To assess the effectiveness of SMAAT against the GCG attack, we conduct the attack by setting the suffix length to 20 tokens. The attack is generated over 50 iterations, with 100 trials per iteration. Rather than leveraging the attack's transferability, we tailor an adversarial suffix for every individual prompt and model to enhance the attack's impact. Our assessment is conducted using two datasets, AdvBench (Zou et al., 2023) and HH-RLHF (Bai et al., 2022), and involves three models: BERT, RoBERTa, and DistilBERT. AdvBench features 640 training prompts (320 harmful, 320 safe) and 240 test prompts (120 harmful, 120 safe), whereas HH-RLHF includes approximately 44K harmful and 44K helpful training prompts, with 2.3K test samples for each prompt category. In all cases, standard models are trained over 5 epochs with a learning rate of $1e^{-5}$. Table 6 details the training hyperparameters for SMAAT.

---

[3] We use the fine-tuned models available from *https://huggingface.co/JeremiahZ*

