# OpenReview forum: "Explaining the role of Intrinsic Dimensionality in Adversarial Training"
_ICML.cc/2025/Conference — ICML 2025 poster_

### Official Review · Reviewer_JtkR · 2025-03-07

**Overall Recommendation:** 3

**Summary:**

This paper investigates the intrinsic dimensionality in the layerwise fashion for adversarially trained models. This paper provides a new perspective for adversarial training in different model architectures from manifold conjecture. The off-manifold adversarial examples (AEs) enhance robustness, and the on-manifold AE improves generalization. From an architectural perspective, decoder-based LLM and commonly used vision models exhibit different characteristics from encoder-based LLM. The vision and decoder-based LLMs exhibit low intrinsic dimensionality in earlier layers (favouring off-manifold AEs), whereas encoder-based models do so in later layers (favouring on-manifold AEs). Based on this property, this paper introduced SMAAT: Scalable Manifold Aware Adversarial Training. Experiments demonstrated its effectiveness and efficiency.

---

## update after rebuttal

After the rebuttal, I have not changed my rating. The authors have addressed my concerns.

**Claims And Evidence:**

Claims made in the submission are supported by convincing evidence.

**Essential References Not Discussed:**

Most of the related works are discussed.

**Experimental Designs Or Analyses:**

The existing experimental designs are technically sound. However, given that SMAAT is more of a generalizable AT framework. It would be beneficial to add evaluation with vision models on typical benchmarks like RobustBench [1]. Additionally, the current evaluation only focused on encoder-based LLMs; it would be more comprehensive to include decoder-based LLMs as well.

[1] Croce, Francesco, et al. "RobustBench: a standardized adversarial robustness benchmark." Thirty-fifth Conference on Neural Information Processing Systems Datasets and Benchmarks Track (Round 2).

**Methods And Evaluation Criteria:**

Proposed methods and evaluation criteria make sense for the problem.

**Other Comments Or Suggestions:**

I suggest discussing the broader benefits of improving adversarial training efficiency, particularly for encoder-based LLMs, as it would strengthen the motivation and demonstrate greater practical significance.

**Other Strengths And Weaknesses:**

Strengths
- The paper provides valuable insights by exploring intrinsic dimensionality in a layer-wise manner across various model architectures, highlighting the important distinctions between off-manifold and on-manifold AEs. These insights could have broad implications and significantly influence future research.
- Reducing the training overhead is a crucial contribution to adversarial training, particularly given the growing preference for larger, more computationally intensive models.

Weaknesses
- The evaluation of text-based models appears limited. Given that SMAAT is presented as a generalized framework, a more comprehensive evaluation, especially on vision models, would significantly strengthen its contribution.
- While the proposed SMAAT framework claims improved efficiency in adversarial training, the current efficiency gains primarily benefit specific architectures (encoder-based LLMs). A broader evaluation across diverse model types, including decoder-based LLMs, vision models, or multimodal architectures, would better demonstrate the generalizability and practical impact of SMAAT.

**Questions For Authors:**

Please address the concerns highlighted in the Experimental Designs and Strengths and Weaknesses sections. Clarifying these points, especially regarding the rigour of the experimental evaluations and the scope of the baseline comparisons, will likely influence my overall assessment of this paper.

**Relation To Broader Scientific Literature:**

The findings of this paper are very interesting and could have the potential for a larger impact on broader scientific literature.

**Theoretical Claims:**

No new theoretical claims.

---

> ### Author Rebuttal · Authors · 2025-03-30
>
> We appreciate the reviewer’s insightful comments and suggestions. In response, we conducted additional experiments to address the raised concerns and outlined the further revisions planned for the paper.
>
> **Evaluation on vision models:**
>
> > Given that SMAAT is presented as a generalized framework, a more comprehensive evaluation, especially on vision models, would significantly strengthen its contribution.
>
> >  It would be beneficial to add evaluation with vision models on typical benchmarks like RobustBench [1].
>
>
> As requested by the reviewer,  we have extended the robustness and generalization analyses of encoder and decoder models to vision models.  Following the approach in Figs. 5 and 6—where one model from each architecture was analyzed—we conducted similar experiments on VGGNet, chosen for its relative computational efficiency. It is important to note that performing a full grid search with adversarial training (AT) across all layers and measuring robustness for each model is highly resource-intensive.
>  Specifically, we performed **AT** on VGGNet using the CIFAR-10 dataset by attacking every ReLU layer over 20 epochs, with **ε = 0.031–0.2** and **lr = 0.01–0.001**, evaluating robustness via **RobustBench**. [Results](https://anonymous.4open.science/r/SMAAT-25-DD9F/ICML%20Rebuttal/vgg_curve.pdf) show that attacking lower layers (light blue) improves robustness but reduces generalization (**bottom-right**), while upper layers (dark blue) enhance generalization at the cost of robustness (**top-left**). This shows that vision models more closely resemble decoder models in terms of their generalization versus robustness characteristics.
>
> **New experiments on decoder-based models:**
>
> > Additionally, the current evaluation only focused on encoder-based LLMs; it would be more comprehensive to include decoder-based LLMs as well.
>
> We conducted new experiments using the PAIR attack on the Llama-2 model.  We performed the robustness versus generalization experiments shown in Fig. 4 under this attack scenario. The corresponding results, for comparison with Fig. 4, can be viewed at [link](https://anonymous.4open.science/r/SMAAT-25-DD9F/ICML%20Rebuttal/llama2_gen_robust_pair.pdf). As seen in the results, we observe the same trend as with the GCG attack. However, while model robustness drops to **30%** under the GCG attack, it remains above **75%** across all setups for the PAIR attack. This further supports our decision to use the GCG attack (i.e., the suffix attack) over PAIR attack in our experiments.
>
> **Efficiency gain assessment:**
>
> >  I suggest discussing the broader benefits of improving adversarial training efficiency, particularly for encoder-based LLMs,
>
> Our analysis of the three most widely used architectures—encoder, decoder, and vision models—reveals a key distinction: encoder-based models uniquely exhibit a decreasing intrinsic dimensionality trend across layers, in contrast to the increasing trend we observed in the other two architectures. The core idea behind our method SMAAT is to apply adversarial training (AT) at the layer with the highest proportion of off-manifold samples to maximize robustness. For encoder models, this corresponds to the last layer, while for decoder and vision models, it aligns with the first layer, effectively making SMAAT equivalent to conventional AT in those cases. In this context, our analysis offers the first explanation for why traditional AT has proven especially effective.
>
> **Emphasizing AT efficiency:**
>
> > I suggest discussing the broader benefits of improving adversarial training efficiency, particularly for encoder-based LLMs
>
> Thank you for this note. Our work fundamentally explores the relationship between layer-wise intrinsic dimensionality (ID) and its effect on the generalization–robustness trade-off.  The proposed SMAAT method leverages these ID-related insights to guide adversarial training. Notably, SMAAT leads to significant improvements for encoder-based models, as highlighted in Fig. 2 of the Introduction. Furthermore, as shown in Table 2, SMAAT introduces no additional overhead beyond standard model training. We will further revise the text to better highlight this aspect.

---

### Official Review · Reviewer_aVnu · 2025-03-10

**Overall Recommendation:** 3

**Summary:**

This paper reveals the fundamental reasons behind the varying effectiveness of adversarial training across different types of neural networks and proposes a novel and efficient training method, SMAAT. The study finds that early layers of vision models (e.g., CNNs) and generative language models (e.g., LLaMA) exhibit low intrinsic dimensionality, making them prone to generating adversarial examples that deviate from the true data distribution (off-manifold samples). This results in adversarial training significantly improving robustness at the cost of generalization. Conversely, in encoder-based language models (e.g., BERT), later layers exhibit low intrinsic dimensionality, causing traditional adversarial training to generate samples closer to the real data distribution (on-manifold samples), preserving generalization but limiting robustness improvements.
Based on this discovery, the authors propose the SMAAT framework, which dynamically selects the network layer with the lowest intrinsic dimensionality for perturbation—applying it to the final layer for encoder models and to the input layer for vision/generative models.
Experiments demonstrate that SMAAT outperforms existing techniques in various applications, including sentiment analysis, content safety filtering, and retrieval-augmented systems.  This study not only provides the first explanation of adversarial training discrepancies from a data distribution perspective but also introduces a new training paradigm that balances efficiency and security. The proposed method is particularly valuable for the rapid deployment of adversarially robust large language models in real-world applications.

**Claims And Evidence:**

I think that the claims presented in this paper are reasonable to some extent and are supported by experimental evidence.

**Essential References Not Discussed:**

There are relatively relevant citations.

**Experimental Designs Or Analyses:**

The paper provides strong evidence for the effectiveness of SMAAT through cross-validation across multiple tasks and attack scenarios. The experimental design generally aligns with field standards, but additional details are needed to further enhance the credibility of the conclusions.

**Methods And Evaluation Criteria:**

The paper proposes the SMAAT method, which selects adversarial training perturbation layers by analyzing the intrinsic dimensionality (ID) of different model layers to enhance robustness and training efficiency. The study covers vision models, encoder-based language models (such as BERT), and decoder-based language models (such as LLaMA) and evaluates the approach across multiple tasks, including text classification, safety filtering, and RAG retrieval.
Overall, the proposed method directly addresses the core challenges of adversarial training in encoder-based models with a well-structured and rigorous evaluation framework. While there are limitations in its applicability to specific scenarios, the approach offers a valuable paradigm for efficient and robust model training.

**Other Comments Or Suggestions:**

Adversarial attacks in the visual domain are well known, and it is worth going into more detail about adversarial attacks in the text domain.

**Other Strengths And Weaknesses:**

The introduction of the paper is not simple and straightforward enough, and some basic concepts are explained in an overly complicated manner.

**Questions For Authors:**

I am unsure about the specific meaning of “data manifold.” If “on-manifold” refers to high-density and “off-manifold” refers to low-density, it may not be necessary to use the term “manifold,” as it could confuse the readers.

**Relation To Broader Scientific Literature:**

This paper integrates the manifold hypothesis, intrinsic dimensionality (ID) analysis, and efficient training methods to construct a theoretically coherent and widely applicable adversarial training framework. Its core innovation lies in revealing the decisive role of layer-wise ID distribution in determining adversarial sample properties and, based on this insight, proposing a scalable perturbation strategy.

**Theoretical Claims:**

The paper primarily presents two theoretical insights: (1) the relationship between intrinsic dimensionality (ID) and the manifold properties of adversarial samples (ONM/OFM); and (2) the effectiveness of the SMAAT method in enhancing robustness and efficiency by perturbing low-ID layers. While the theoretical claims are strongly supported by systematic experimental design—spanning different models, tasks, and attack scenarios—the paper lacks rigorous mathematical proof.

---

> ### Author Rebuttal · Authors · 2025-03-30
>
> We thank the reviewer for their helpful comments and suggestions aimed at improving the clarity of the paper. Below are our responses to the reviewer’s points.
>
>
> **Theoretical proofs:**
>
> > While the theoretical claims are strongly supported by systematic experimental design—spanning different models, tasks, and attack scenarios—the paper lacks rigorous mathematical proof.
>
> The paper relies on two statements:  (1) AEs generated from a low-dimensional manifold are likely to be off-manifold (similarly, AEs generated from a high-dimensional manifold are likely to be on-manifold) and (2) the manifold conjecture stating that off/on-manifold AEs lead to better robustness/generalization. While the manifold conjecture is well established in the literature (Ethayarajh, 2019; Shamir et al., 2021; Gilmer et al., 2018), statement (2) is straightforward:  Let $\mathcal{M} \subset \mathbb{R}^n$ be a smooth, compact, low-dimensional manifold with intrinsic dimension $d \ll n$, embedded in $\mathbb{R}^n$. Let $f: \mathbb{R}^n \to \mathbb{R}^k$ be a classifier trained on data sampled from $\mathcal{M}$. Let $\delta$ denote an adversarial perturbation obtained by maximizing the loss $\mathcal{L}(f(x + \delta), y)$ under a norm constraint $\|\delta\| \leq \epsilon$. Since the classifier models $p(y \mid x)$ without access to the generative distribution $p(x)$, it lacks explicit knowledge of the manifold $\mathcal{M}$. Therefore, the loss gradient $\nabla_{\delta} \mathcal{L}$ generally has components orthogonal to the tangent space. As a result, the perturbation $\delta$ is unlikely to lie entirely within $\mathcal{M}$, and typically has a non-zero orthogonal component. Moreover, the smaller the dimension $d$ of the manifold relative to $n$, the larger the proportion of $\delta$ that lies off-manifold. Hence, adversarial examples generated in this way are very likely to lie off the data manifold. We will add this explanation to the paper. We hope this clarifies.
>
> **Clarity of the introduction:**
>
> > The introduction of the paper is not simple and straightforward enough, and some basic concepts are explained in an overly complicated manner.
>
> Thank you for your comment. Below, we clarify the information presented in the introduction. Moreover, we are happy to accommodate any specific suggestions you may have to improve it further.
> Currently, our Introduction is structured as follows: (A) We begin by introducing adversarial training (AT) and highlighting two major limitations that our work addresses: (1) the poorly understood tradeoff between robustness and generalization (L.51–52, Col.1), and (2) the high computational cost of AT, which limits its practical deployment (L.51–52, Col.2). (B) To address the first limitation, we investigate how encoder LLMs, vision models, and decoder LLMs differ in intrinsic dimensionality, leading to distinct compositions of on-/off-manifold adversarial examples. This, under the manifold conjecture, explains their varying impacts on robustness and generalization. (C) To address the second limitation, we propose a scalable Manifold-Aware Adversarial Training approach that selectively applies AT at the layer with the highest proportion of off-manifold AEs, significantly reducing cost without sacrificing performance.
>  (D) Finally, we summarize our experimental results supporting both contributions. We hope this clarifies the structure and motivation of the introduction.
>
> **Usage of the term manifold**
>
> > I am unsure about the specific meaning of “data manifold.” If “on-manifold” refers to high-density and “off-manifold” refers to low-density, it may not be necessary to use the term “manifold,” as it could confuse the readers.
>
> We formally define the ‘data manifold’ and ‘on-/off-manifold samples’ in Section 3. To improve clarity, we have revised the introduction to introduce these definitions earlier. Specifically, we use the term data manifold to refer to a potential non-linear subspace spanned by the dataset as it propagates through the network layers, with its dimension quantified via a projection-based method.  In this context, on-/off-manifold samples refer to data points that are either captured by or fall outside the learned manifold during training.
>
>
> **Adversarial attacks in the text domain**
>
> > Adversarial attacks in the visual domain are well known, and it is worth going into more detail about adversarial attacks in the text domain.
>
> We will revise the introduction and related work to better describe adversarial attacks on text, with an emphasis on most effective ones such as GCG attack, i.e., the suffix attack, and PAIR attack.

---

### Official Review · Reviewer_fFMr · 2025-03-14

**Overall Recommendation:** 4

**Summary:**

The authors investigate how the relationship between perturbations and the data manifold influences whether adversarial training leads to improved generalization or robustness. Based on this insight, they propose SMAAT, a method that generates perturbations at specific layers to target different manifolds—leveraging the fact that intrinsic dimensionality changes with layer depth—to achieve more precise trade-offs between generalization and robustness. The paper provides extensive quantitative results on LLMs supporting these claims, demonstrating that SMAAT is more efficient and achieves a better generalization-robustness tradeoff compared to existing methods.

**Claims And Evidence:**

Yes.

**Essential References Not Discussed:**

No.

**Experimental Designs Or Analyses:**

Figure 1 and Figure 2 lack sufficient experimental details. It would be helpful to explicitly clarify these details in the experiments section or provide additional explanations in the supplementary material.

**Methods And Evaluation Criteria:**

Yes.

**Other Comments Or Suggestions:**

- define terms when first used and in important locations (e.g. RAG in abstract, LAT in figure 1)
- figure 1: could improve clarity by just plotting points and adding best-fit lines to show trends more clearly

**Other Strengths And Weaknesses:**

Strengths:
- Clear writing with tightly integrated experiments, making the claims and findings easy to follow.
- Strong and wide-ranging experiments, demonstrating both efficiency and a well-balanced generalization-robustness tradeoff.
- Relatively simple yet effective method that directly leverages the observed manifold-robustness-generalization relationship, reinforcing its validity and practical significance.

Weaknesses:
- Weak/easy adversarial attack used in Table 3: Evaluating accuracy under attack against more recent and stronger adversarial attacks (e.g., [2,3]) would better demonstrate the method’s robustness improvements.
- Limited experiments on vision models: While Figure 4 explores intrinsic dimensionality and reconstruction error for vision models, the paper does not provide robustness-generalization results for vision models using the proposed method. Since Figure 3 suggests the method can enhance vision model robustness, a direct comparison on vision benchmarks would strengthen the findings.

references:

[1] Zhang, Hongyang, et al. "Theoretically principled trade-off between robustness and accuracy." International conference on machine learning. PMLR, 2019.

[2] Liu, Xiaogeng, et al. "Autodan: Generating stealthy jailbreak prompts on aligned large language models." arXiv preprint arXiv:2310.04451 (2023).

[3] Chao, Patrick, et al. "Jailbreaking black box large language models in twenty queries." arXiv preprint arXiv:2310.08419 (2023).

**Questions For Authors:**

Q1. While I understand that evaluating vision model robustness requires significantly more experimental setup than adding another text-based experiment, is there a specific reason why no experiments were conducted on vision models using the proposed method?

Q2. Do the authors plan on releasing the code for their experiments?

**Relation To Broader Scientific Literature:**

This work builds on previous research on adversarial training, particularly the accuracy-robustness tradeoff in image models [1], and extends these insights to generalization properties and different architectures. The empirical observation of a strong relationship between the data manifold and the robustness-generalization tradeoff is a valuable contribution, as it provides a deeper understanding that can inform more refined adversarial training strategies.

**Theoretical Claims:**

N/A

---

> ### Author Rebuttal · Authors · 2025-03-30
>
> We thank the reviewer for their valuable comments and suggestions. In response, we have conducted additional experiments to address the critiques and provide a summary of further revisions that will be made to the paper.
>
> **Details on Fig. 1 and Fig. 2:**
> > Figure 1 and Figure 2 lack sufficient experimental details ...
>
> These two figures summarize key findings presented in Sec. 4 and Sec.  5.
>
>
> Figure 1 presents results on the LLaMA-2 model by combining the ID characteristics from Fig. 4(d) (1st row), reconstruction error from Fig. 4(d) (2nd row), and the generalization and robustness trends from Fig. 5. For Fig. 1, we report the best results that maximize the sum of robustness and generalization scores for each layer. Additional details are provided in the supplementary material.
>
>
> Figure 2 compares  SMAAT against other  AT approaches for  encoder-based models, evaluating generalization, robustness, and run-time cost across different tasks.
>
>
> We will revise the Introduction to provide further details and to reference the Supplementary for full details on the generation of Figs. 1 and 2.
>
>
> **Additional attacks:**
> > Evaluating accuracy under attack against more recent and stronger adversarial attacks (e.g., [2,3]) would better demonstrate the method’s robustness improvements.
>
> Although we acknowledge that the suffix attack (the method used in Table 3) is not a recent technique, it remains one of the most powerful attack strategies. For the sake of clarity, we conducted new experiments using the PAIR attack, in addition to the suffix attack. Since Table 3 reports results for encoder-based models—which are not compatible with attacks that rely on generative capabilities like PAIR—we instead applied this attack to the LLaMA-2 model. We performed the robustness versus generalization experiments shown in Fig. 4 under this attack scenario. The corresponding results, for comparison with Fig. 4, can be viewed at [link](https://anonymous.4open.science/r/SMAAT-25-DD9F/ICML%20Rebuttal/llama2_gen_robust_pair.pdf). As seen in the results, we observe the same trend as with the GCG (i.e., the suffix attack) attack. However, while model robustness drops to **30%** under the GCG attack, it remains above **75%** across all setups for the PAIR attack. This further supports our decision to use the GCG attack over PAIR in our experiments.
>
>
> **Experiments on vision models:**
>
> > Q1. While I understand that evaluating vision model robustness requires significantly more experimental setup than adding another text-based experiment, is there a specific reason why no experiments were conducted on vision models using the proposed method?
>
> > While Figure 4 explores intrinsic dimensionality and reconstruction error for vision models, the paper does not provide robustness-generalization results for vision models using the proposed method.
>
> We extend the robustness and generalization analyses of the encoder and decoder models to vision models. Following the approach in Figs. 5 and 6—where one model from each architecture was analyzed—we conducted similar experiments on VGGNet, chosen for its relative computational efficiency. It is important to note that performing a full grid search with adversarial training (AT) across all layers and measuring robustness for each model is highly resource-intensive.
> Specifically, we performed  **AT** on VGGNet using the CIFAR-10 dataset by attacking every ReLU layer over 20 epochs, with **ε = 0.031–0.2** and **lr = 0.01–0.001**, evaluating robustness via **RobustBench**. [Results](https://anonymous.4open.science/r/SMAAT-25-DD9F/ICML%20Rebuttal/vgg_curve.pdf) show that attacking lower layers (light blue) improves robustness but reduces generalization (**bottom-right**), while upper layers (dark blue) enhance generalization at the cost of robustness (**top-left**). This shows that vision models more closely resemble decoder models in terms of their generalization versus robustness characteristics.
>
>
> **Abbreviations:**
> >  define terms when first used and in important locations (e.g. RAG in abstract, LAT in figure 1)
>
> We will revise the text to define terms where they appear first.
>
>
> **Improving clarity of Fig. 1:**
>
> > figure 1: could improve clarity by just plotting points and adding best-fit lines to show trends more clearly
>
> Thanks. We have regenerated Fig. 1 to display best-fit lines using a locally weighted regression technique. The updated figure is available at the [link](https://anonymous.4open.science/r/SMAAT-25-DD9F/ICML%20Rebuttal/llama_all_in_one_trend.pdf).
>
>
> **Code:**
>
> > Q2. Do the authors plan on releasing the code for their experiments?
>
> The code is publicly available and can be accessed at the [link](https://anonymous.4open.science/r/SMAAT-25-DD9F/README.md)

---

### Decision · Program_Chairs · 2025-05-01

**Decision:**

Accept (poster)

**Comment:**

This paper studies how the interaction between adversarial perturbations and the data manifold influences whether adversarial training boosts generalization or robustness. Based on this insight, the authors propose to strategically apply perturbations at specific layers (perturbing later layers of encoder-based models and earlier layers of vision/decoder models). This approach shows promising results for encoder-based adversarial training. All reviewers agree that this is a good paper; the insight is clear, and the results are promising. Some detailed questions (mostly related to experiments, clarity, and code) were addressed during the rebuttal phase. We thus recommend acceptance of the paper.